# Characterization of the Microbial Communities along the Gastrointestinal Tract in Crossbred Cattle

**DOI:** 10.3390/ani12070825

**Published:** 2022-03-24

**Authors:** Kai Wang, Hailiang Zhang, Lirong Hu, Guoxing Zhang, Haibo Lu, Hanpeng Luo, Shanjiang Zhao, Huabin Zhu, Yachun Wang

**Affiliations:** 1College of Animal Science and Technology, China Agricultural University, Beijing 100193, China; kai@cau.edu.cn (K.W.); zhl108@cau.edu.cn (H.Z.); b20193040324@cau.edu.cn (L.H.); wledu2016@163.com (G.Z.); luohanpeng@cau.edu.cn (H.L.); 2Beijing SUNLON Biological Seed Industry Innovation Technology Limited Company, Beijing 101206, China; luhaibo979@163.com; 3Institute of Animal Sciences, Chinese Academy of Agricultural Sciences, Beijing 100193, China; zhaoshanjiang@caas.cn

**Keywords:** gut microbiota, 16s rRNA gene sequencing, crossbred cattle, functional prediction, Simmental × Holstein crossbred

## Abstract

**Simple Summary:**

Crossbreeding has been used worldwide to improve milk production, milk composition, and reproduction performance. Understanding the structure of the microbial communities in the gastrointestinal tract (GIT) of crossbred cattle is paramount for developing new livestock management technologies with an emphasis on nutrition and sustainability. In this study, we investigated the gastrointestinal microbiota of Simmental × Holstein crossbred cattle using 16s rRNA gene sequencing. Microbial communities in the small intestine had the lowest diversity of bacteria and highest diversity of bacterial functions, and three groups of GIT regions, including the stomach, small intestine, and large intestine were characterized by specific bacteria and bacterial functions. In summary, spatial heterogeneity of the microbiota was found across the GIT of crossbreeds, and specific microbial biomarkers were identified in different regions.

**Abstract:**

The gastrointestinal microbiota greatly affects the health status and production performance of bovines. Presently, many studies have used high-throughput sequencing methods to investigate the gastrointestinal microbiome in bovines. However, the microbiome profile of crossbred cattle across the whole gastrointestinal tract (GIT) has not been thoroughly reported. In this study, the digesta at ten regions (including the rumen, reticulum, omasum, abomasum, duodenum, jejunum, ileum, cecum, colon, and rectum) of the GIT were collected in three Simmental × Holstein crossbred heifers aged 17 months, and microbial DNA was extracted and amplified for sequencing of the V3–V4 regions of the 16S rRNA gene. Functional orthologs of the microbiota genome were predicted and analyzed. We found that samples were categorized into three groups (the stomach, small intestine, and large intestine) by principal coordinate analysis (PCoA) based on Bray–Curtis dissimilarity in both the bacterial composition and functional profile. Samples from small intestine had the lowest alpha diversity of bacteria composition and highest alpha diversity of the functional composition. Three groups of GIT regions were characterized by several microbiome features. The stomach was characterized by *Bacteroidetes* and *Fibrobacteres* at the phylum level, and KEGG pathways related to the metabolism of cofactors and vitamins, glycan biosynthesis, and metabolism were enriched in the stomach. The small intestine was characterized by *Actinobacteria* and *Patescibacteria* at the phylum level, and KEGG pathways related to xenobiotics biodegradation and metabolism were enriched in the small intestine. The large intestine featured *Ruminococcaceae*, *Rikenellaceae*, and *Bacteroidacea* at the family level, and KEGG pathways, including steroid hormone biosynthesis, linoleic acid metabolism, and cysteine and methionine metabolism were enriched in the large intestine. The results of the current study revealed the spatial heterogeneity of microbiota across the GIT in Simmental × Holstein crossbreeds and identified microbial biomarkers of different regions. The results can provide useful information for the study of the gastrointestinal microbiome in bovines.

## 1. Introduction

The microbes in the gastrointestinal tract (GIT) have long been recognized as essential factors in the process of digesting nutrients and for host health [1]. In particular, bacteria in the bovine gut play a major role in the biological degradation of dietary fibers, non-protein nitrogen, and other nutrients. In bovines, feedstuffs are processed and converted into volatile fatty acids, microbial protein, and vitamins by gastrointestinal bacteria to meet the requirements for the maintenance, growth, production, and health of the host [2,3].

Cattle also use gastrointestinal bacteria for the development of the immune system of the GIT [4,5]. Due to the above-mentioned indispensable roles, illustrating the diversity of the entire microbial communities in the GIT is of great significance for bovines. 16s rRNA gene sequencing offers a method for a deeper appreciation of the diversity and composition of gastrointestinal microbiota, and molecular functions of the gastrointestinal microbiome can be inferred by bioinformatic tools, such as PICTUSt2 [6].

In the previous decades, crossbreeding has been used worldwide to improve milk production, milk composition, and reproduction performance [7]. The Simmental and Holstein crossbreeding is one of the most common crossbreeding patterns, which has shown efficient herd improvement and reduced inbreeding coefficients in several studies [8,9,10]. Understanding the structure of the microbial community of these crossbred cattle may be useful for developing new management technologies by regulating the gut microbiome, with an emphasis on nutrition and sustainability.

To date, many studies have used high-throughput sequencing of the 16S rRNA gene to investigate the rumen and feces microbiota in bovines, whereas other regions of the GIT, such as the abomasum, small intestine, and cecum, have not been thoroughly reported [11,12,13,14,15,16,17,18,19,20,21]. Limited studies have explored the microbial communities across the whole GIT in Chinese Holstein cattle [16], Brazilian Nelore cattle [19], Xuanhan yellow cattle (XHC), and Simmental × XHC crossbred cattle [20].

However, there is no literature investigating microbial communities across the whole GIT in crossbred cattle, and most of the above-mentioned studies have not investigated the functions of the gastrointestinal microbiome. Therefore, the objective of this study is to clarify the taxonomic and functional profile of the microbiome at ten regions across the whole GIT (rumen, reticulum, omasum, abomasum, duodenum, jejunum, ileum, cecum, colon, and rectum) in Simmental × Holstein crossbred cattle and to provide useful information for the significance of the gastrointestinal microbiome in crossbred cattle.

## 2. Materials and Methods

### 2.1. Animals and Sample Collection

All animal procedures used in this study were approved by the Animal Care and Use Committee of the Institute of Animal Sciences of Chinese Academy of Agricultural Sciences (IAS/CAAS). The experiment was conducted at the experimental farm of IAS/CAAS. Three Simmental × Holstein crossbred heifers reared at the same barn were selected and fed ad libitum with a corn-soybean-based diet, which was formulated to meet the nutritional requirements of the animals. Animals were aged 17 months and weighted 360 kg on average at slaughter.

After slaughter, ten segments of each GIT were immediately separated from the mesentery using surgical knife, including four segments of the stomach (rumen, reticulum, omasum, and abomasum), three segments of the small intestine (duodenum, jejunum, and ileum), and three segments of the large intestine (cecum, colon, and rectum). Subsequently, each segment was placed on a sterilized plate and transferred to the laboratory. A plate can only be used for one GIT segment. The gastrointestinal contents in the middle of each segment were homogenized and sampled into a 5 mL sterile tube using sterile medical gloves. Medical gloves were discarded after completing the previous sample, and new ones were used for the new sample. All samples were immediately placed on carbon dioxide before they were stored at −80 °C in the laboratory until sequencing. A total of 29 samples were sequenced for downward analysis (one sample was excluded because of an error in labelling).

### 2.2. DNA Extraction and 16S rRNA Gene Sequencing

Microbial DNA was extracted from samples using the E.Z.N.A.^®^ soil DNA Kit (Omega Bio-tek, Norcross, GA, USA) according to manufacturer’s protocols. The final DNA concentration and purification were determined using a NanoDrop 2000 UV-vis spectrophotometer (Thermo Scientific, Wilmington, SC, USA), and the DNA quality was checked via 1% agarose gel electrophoresis. The V3–V4 hypervariable regions of the 16S rRNA genes were amplified with the primers 338F (5′-ACTCCTACGGGAGGCAGCAG-3′) and 806R (5′-GGACTACHVGGGTWTCTAAT-3′) by a thermocycler PCR system (GeneAmp 9700, ABI, New York, NY, USA).

The PCR reactions were conducted using the following program: 3 min of denaturation at 95 °C, 27 cycles of 30 s at 95 °C, 30 s for annealing at 55 °C, 45 s for elongation at 72 °C, and a final extension at 72 °C for 10 min. PCR reactions were performed in triplicate 20 μL mixture containing 4 μL of 5 × FastPfu Buffer, 2 μL of 2.5 mM dNTPs, 0.8 μL of each primer (5 μM), 0.4 μL of FastPfu Polymerase, and 10 ng of template DNA. The resulting PCR products were extracted from a 2% agarose gel, further purified using the AxyPrep DNA Gel Extraction Kit (Axygen Biosciences, Union City, CA, USA), and quantified using QuantiFluor™-ST (Promega, WI, USA) according to the manufacturer’s protocol. Purified amplicons were pooled in equimolar and paired-end sequenced (2 × 300 bp) on an Illumina MiSeq platform (Illumina, San Diego, CA, USA) according to the standard protocols.

### 2.3. Sequence Data Processing

The QIIME2 platform [22] was used to identify amplicon sequence variants (ASVs) from the sequence data, assign taxonomic information for each ASV, and calculate community diversity indices. Default parameters were used in the q2-data2 plugin to filter, denoise, and merge paired-end reads except for trimming off the barcode (6 nt in length) and primer sequence (20 nt in length). A total of 843,763 high-quality reads were retained and mapped to 6540 unique ASVs.

Singletons (ASVs found only in one sample) were removed for statistical analysis, resulting in 4154 high quality ASVs. A Naive Bayes classifier was then trained based on the SILVA reference database (release 132) [23] (reference reads were trimmed to the V3–V4 region bound by the 338F/806R primer pair) using the q2-feature-classifier plugin and used to assign taxonomic information for each ASV. The microbiota alpha diversity (Richness, Shannon) and beta diversity (Bray–Curtis) indices were calculated using q2-diversity plugin. When calculating diversity indices, the total frequency of each sample was rarefied to 19,225, which was the minimum total frequency of all samples.

PICRUSt2 [6] was used to predict functional profile (i.e., KEGG ortholog (KO) abundances) of microbiota in this study. The alpha diversity (including Richness and Shannon index) of functional profile in each sample, and Bray–Curtis distance in the predicted composition of KOs among samples were calculated using the vegan package [24] in R. The abundance of a third-level KEGG pathway was calculated by summing the abundances of KOs in the same pathway.

### 2.4. Statistical Analysis

The significance of comparisons of the microbiota alpha diversity among the stomach, small intestine, and large intestine were determined using the Wilcoxon rank sum test in R (version 4.1.0). Measurements of beta diversity were determined using principal coordinate analysis (PCoA), and the significance of clustering was determined by PERMANOVA with 999 permutations using the vegan package [24] in R. Microbial biomarkers, including bacterial taxa from the phylum to species level and KEGG pathways at the third level, were identified using the Linear effect size (LEfSe) analysis in the MicrobiotaProcess package [25] in R. The threshold values of the false discovery rate (FDR) and logarithmic LDA score were set to 0.01 and 3, respectively.

## 3. Results

### 3.1. Spatial Change of Microbial Community Diversity across GIT

Changing patterns of alpha diversity across GIT were different between microbiota bacteria composition and the predicted functional profile. Samples from the small intestine had the significantly lowest richness and Shannon index of the composition of ASVs (Figure 1A), whereas the alpha diversity based on the composition of KOs was significantly higher in the small intestine compared with in the stomach and large intestine (Figure 1B). There was no significant difference between stomach and large intestine based on both the composition of ASVs and KOs (Figure 1A,B).

Ordination of the Bray–Curtis dissimilarity by PCoA revealed the separation of three groups of samples corresponding to the stomach, small intestine, and large intestine (Figure 1C,D), and the dissimilarity of the bacteria composition and predicted functional profiles between three groups were significant (PERMANOVA, *p* < 0.01).

As shown in a Venn diagram, the stomach, small intestine, and large intestine shared a very small amount of ASVs (1.7% of all unique ASVs) (Figure 1E), but a large number of KOs (70.9% of all unique KOs) were shared among the three groups (Figure 1F).

### 3.2. Taxonomic Composition of the Microbiota in GIT

The taxonomic information of each ASV was assigned based on the SILVA reference database (release 132) [23]. ASVs were assigned to 20 phyla, 32 classes, 56 orders, 82 families, 207 genera and 74 species. The proportion of unassigned ASVs ranged from 96.68% (species) to 0.05% (phylum) at each level.

At the phylum level (Figure 2A), the most dominant bacterial phyla in the stomach and large intestine were *Firmicutes* and *Bacteroidetes*. *Firmicutes* was also the most dominant phyla in the small intestine, whereas *Bacteroidetes* was scarcely found in the small intestine. At the family level (Figure 2B), *Prevotellaceae*, *Peptostreptococcaceae,* and *Rumenococcus* were the most predominant bacteria in the stomach, small intestine, and large intestine, respectively. Moreover, *Ruminococcaceae*, together with *Lachnospiraceae* and *Christensenellaceae,* were found in all regions of the GIT with considerable abundance.

### 3.3. Microbial Biomarkers of the Stomach, Small Intestine, and Large Intestine

In total, 111 bacterial taxa from phylum to species level were identified as characteristic bacteria of different regions of the GIT, from which 67 bacteria with considerable abundance (with a mean relative abundance larger than 1% in at least one group) are displayed in the Cladogram (Figure 3). At the phylum level, the stomach was mainly characterized by *Bacteroidetes* and *Fibrobacteres*, whereas *Actinobacteria* and *Patescibacteria* were identified as the biomarkers for the small intestine.

At the family level, all differentially abundant taxa from *Bacteroidetes* except *Rikenellaceae* and *Bacteroidaceae* were biomarkers for the stomach, such as *Prevotellaceae* and *Muribaculaceae*. *Ruminococcaceae*, together with *Rikenellaceae* and *Bacteroidaceae*, were characteristically abundant in the large intestine. However, some genera from *Ruminococcaceae*, such as *Ruminococcaceae UCG−011* and *Ruminococcus,* were biomarkers of the stomach. The small intestine featured all differential families from *Actinobacteria* and *Patescibacteria*, including *Eggerthellaceae*, *Atopobiaceae,* and *Nocardiaceae* from *Actinobacteria* as well as *Saccharimonadaceae* from *Patescibacteria*.

A total of 104 third-level KEGG pathways belonging to six first level categories were identified as differential characteristics in GIT, including 60 pathways in the metabolism category and 44 pathways in other first level categories (14 organismal systems, 12 human diseases, seven in environmental information processing, seven cellular processes, and four in genetic information processing).

For pathways related to the metabolism, all differentially abundant pathways related to the metabolism of cofactors and vitamins (annotated with pink in the left bar, Figure 4A), and glycan biosynthesis and metabolism (annotated with red in the left bar, Figure 4A) were enriched in the stomach, whereas pathways related to xenobiotics biodegradation and metabolism (annotated with blue color in the left bar, Figure 4A) were enriched in the small intestine. The large intestine was characterized by steroid hormone biosynthesis, linoleic acid metabolism, and cysteine and methionine metabolism. For pathways in other first level categories, ABC transporters, two-component systems, quorum sensing, and the phosphotransferase system (PTS), cysteine and methionine metabolism had the highest logarithmic LDA score and were enriched in the small intestine (Figure 4B).

## 4. Discussion

The present study thoroughly investigated the microbial community in the GIT of Simmental × Holstein crossbred cattle from two aspects: the bacterial composition and functional profile.

For the overall structure of the bacterial community, our results of microbiota alpha diversity revealed that the stomach and large intestine had a higher richness of bacterial types compared with the small intestine, which is in accordance with studies conducted in milking Holstein cows [16] and Nelore steer [19]. Moreover, our research found that microbiota in the small intestine had the most diverse molecular functions compared with the stomach and large intestine, which suggests that relatively few bacterial types in the small intestine perform diverse functions. The variation of microbial communities at different regions of the GIT confirmed the spatial heterogeneity of the microbiota along the GIT, which may be a result of intestine physiological factors, such as oxygen gradients [26], pH levels [27], and nutrient availability [28]. In our study, the dissimilarity of the microbiota composition between the stomach, small intestine, and large intestine was determined using PCoA and PERMANOVA, and there was a certain degree of similarity between samples in the same group, which is similar to the results from other studies [16,18,19], proving that microbiota in the bovine’s GIT could be grouped into three groups (the stomach, small intestine, and large intestine).

In our study, the phylum *Firmicutes* was dominant across the entire GIT. Among *Firmicutes*, *Ruminococcaceae*, *Lachnospiraceae,* and *Christensenellaceae* were the dominant bacteria families along the entire GIT—a finding that is consistent with that of a study conducted in sheep [18], indicating that these bacteria can survive in diverse conditions and may exert fundamental functions in GIT. *Lachnospiraceae* includes species of bacteria with fibrolytic and proteolytic properties [13], while most of the *Ruminococcaceae* are known to be major degraders of resistant polysaccharides and to provide a range of degradative enzyme systems that enable the host to break up plant cell walls [18].

However, many bacteria occurred with differential abundances along the GIT, and different regions of the GIT can be distinguished by several bacteria, which may exert different physiological functions. *Bacteroidetes* was mainly found in the stomach and large intestine in our study as well as in previous studies [16,19]. Several bacteria of *Bacteroidetes* are considered primary degraders of polysaccharides [29], including taxa from family *Prevotellaceae* [30,31], which was identified as a stomach biomarker in our study. In addition to saccharolytic function, some bacteria of *Bacteroidetes* also have the ability to digest fat and protein, including the genus *Alistipes* from the family *Rikenellaceae* and genus *Bacteroides* from the family *Bacteroidaceae* [32,33], which were identified as large intestine biomarker in our study.

The phylum *Fibrobacteres* was mainly found in the stomach, and its ability to degrade plant-based cellulose has been previously confirmed [34,35]. It has been reported that bacteria from the phylum *Actinobacteria*, such as *Eggerthellaceae* and *Nocardiaceae*, can produce bioactive metabolites, and these produce about two-thirds of all naturally derived antibiotics in current clinical use [36,37]. *Eggerthellaceae* and *Nocardiaceae* were identified as small intestine biomarkers in our study, and they may play important roles in the homeostasis of the intestine. Although many characteristic bacteria were identified in this study, a large proportion of bacteria was not classified at a more precise taxonomic level, such as at the species level by 16s rRNA based on the existing reference database. Hence, research on the more precise taxonomic level is still required in the future.

The cattle gastrointestinal microbiome exerts many physiological functions lacking in the host [16], and therefore these bacteria can be considered essential to cattle life. However, many bacteria have not been cultured in vitro due to restricted requirements. By inferring the genome of all bacteria in an environment, high-throughput sequencing provides an effective way to investigate the genome functions of gut microbiota [6].

In this study, the abundance of functional orthologs in the gastrointestinal microbiome were inferred based on the KEGG database and were further used to infer the abundance of pathways. Differential analysis revealed that a large amount of differentially abundant pathways were related to the metabolism, which was consistent with a previous study in Holstein cattle [16]. Metabolism pathways are considered essential for bacteria survival [38,39], and different GIT regions were characterized by different pathways in the metabolism category in our study.

The stomach was characterized by glycan biosynthesis and metabolism as well as the metabolism of cofactors and vitamins, which further confirmed that certain bacteria play important roles in converting nutrients in the stomach. Microbiota in the small intestine also had important roles in converting nutrients, which was suggested by the enrichment of the lipid metabolism, such as fatty acid degradation, primary bile acid biosynthesis, and ether lipid metabolism. Moreover, the microbiota in the small intestine featured xenobiotics biodegradation and metabolism, such as steroid degradation and xylene degradation, suggesting important roles in host health. Based on the study of predicted functions of the gastrointestinal microbiota, there is a need to investigate the molecular functions of microbiota using more precise methods, such as metagenome sequencing, metatranscriptome sequencing, and metabolomics.

## 5. Conclusions

The current study revealed the spatial heterogeneity of the microbiota across the GIT in the Simmental × Holstein crossbreed. The microbiota were categorized into three groups (the stomach, small intestine, and large intestine) based on dissimilarity in both the bacteria composition and functional profile. Each group had specific characteristics. The microbiota in the small intestine had the least diverse bacteria types and most diverse functions.

At the phylum level, the stomach was mainly characterized by *Bacteroidetes* and *Fibrobacteres*, whereas the small intestine featured *Actinobacteria* and *Patescibacteria*. The large intestine featured *Ruminococcaceae*, *Rikenellaceae,* and *Bacteroidaceae* at the family level. For the KEGG pathways, all differentially abundant pathways related to the metabolism of cofactors and vitamins, and glycan biosynthesis and metabolism were enriched in the stomach, whereas pathways related to xenobiotics biodegradation and the metabolism were enriched in the small intestine. The large intestine was characterized by steroid hormone biosynthesis, linoleic acid metabolism, and cysteine and methionine metabolism.

This study fills in gaps in the gastrointestinal microbiome knowledge of crossbred cattle and can provide useful information for the study of the gastrointestinal microbiome in bovines. However, limitations existing in the present study, such as the small sample size, are acknowledged.

## Figures and Tables

**Figure 1 animals-12-00825-f001:**
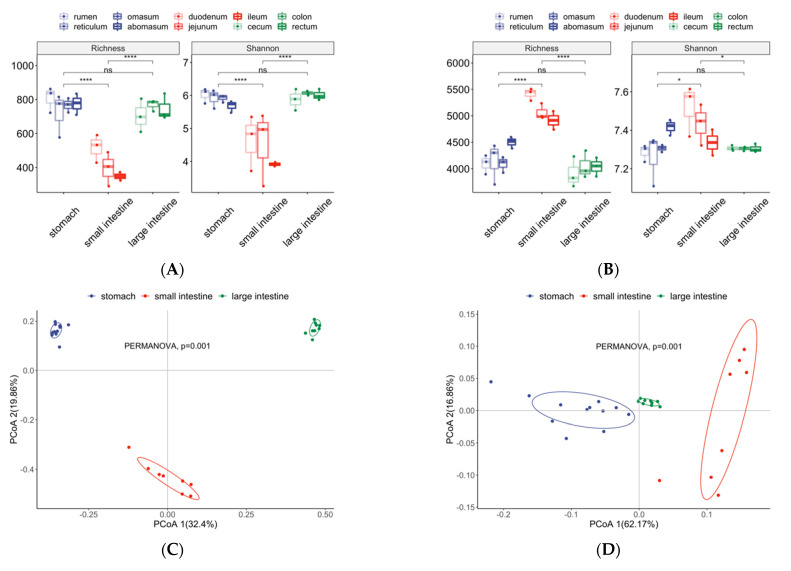
Differences in the microbial community diversity among different regions across the gastrointestinal tract (GIT). (**A**,**B**) Counts and percentage of unique amplicon sequence variants (ASVs) (**A**) and KOs (**B**) in the stomach, small intestine, and large intestine as well as intersections between groups. (**C**,**D**) Comparisons of community alpha diversity based on the composition of ASVs (**C**) and KOs (**D**) between the stomach, small intestine, and large intestine. The Wilcoxon rank sum test was performed to test the significance of pairwise comparisons, and *p*-values were coded as **** (*p* < 0.0001), * (*p* < 0.05), and ns (*p* > 0.05). (**E**,**F**) Principal coordinate analysis (PCoA) based on the Bray–Curtis distance was used to visualize dissimilarity in the community composition of ASVs (**E**) and KOs (**F**) among different samples. The significance of differences among the stomach, small intestine, and large intestine was determined using permutational multivariate ANOVA (PERMANOVA).

**Figure 2 animals-12-00825-f002:**
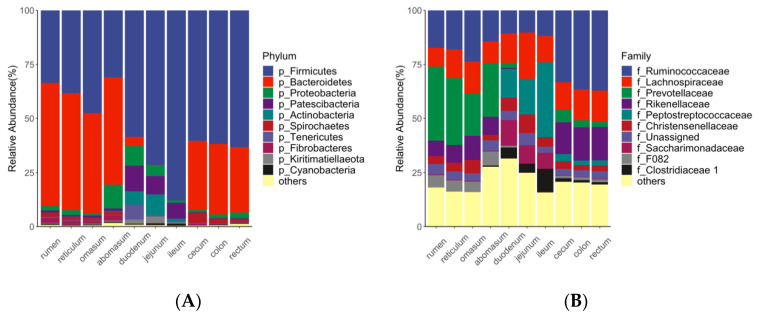
The taxonomic profile of microbial communities at the phylum (**A**) and family level (**B**) across different regions of gastrointestinal tract (GIT). Taxa with relative abundance ranked below 10 were grouped in others.

**Figure 3 animals-12-00825-f003:**
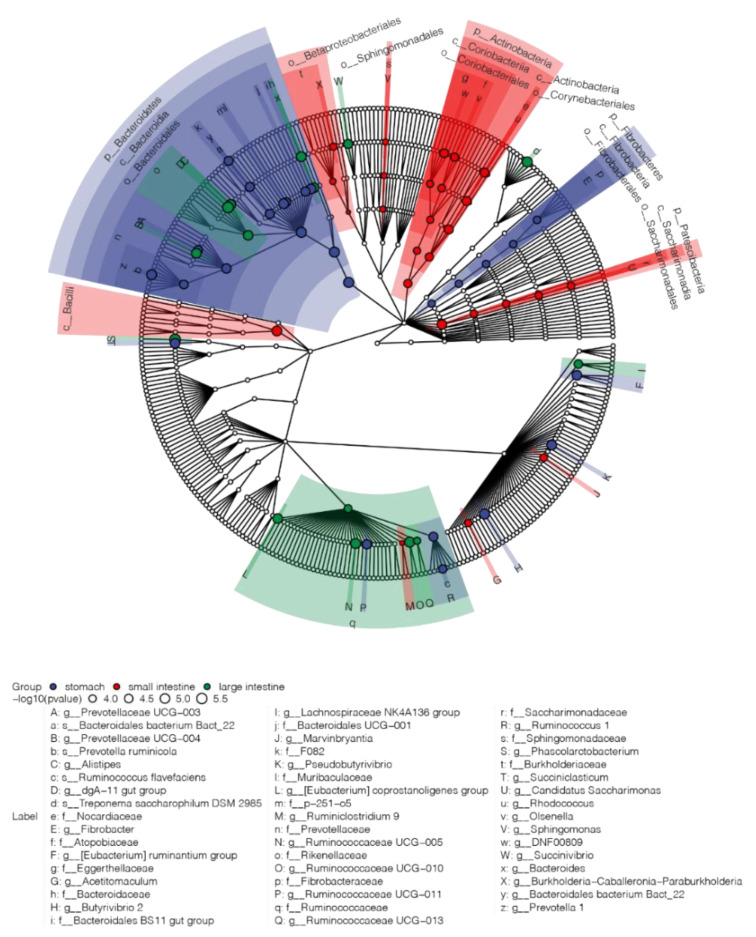
Cladogram of the characteristic bacteria of the stomach, small intestine, and large intestine. Only characteristic bacteria with a relative abundance higher than 1% in at least one group are displayed in cladogram, and bacteria were colored with corresponding colors (stomach: blue; small intestine: red; and large intestine: green).

**Figure 4 animals-12-00825-f004:**
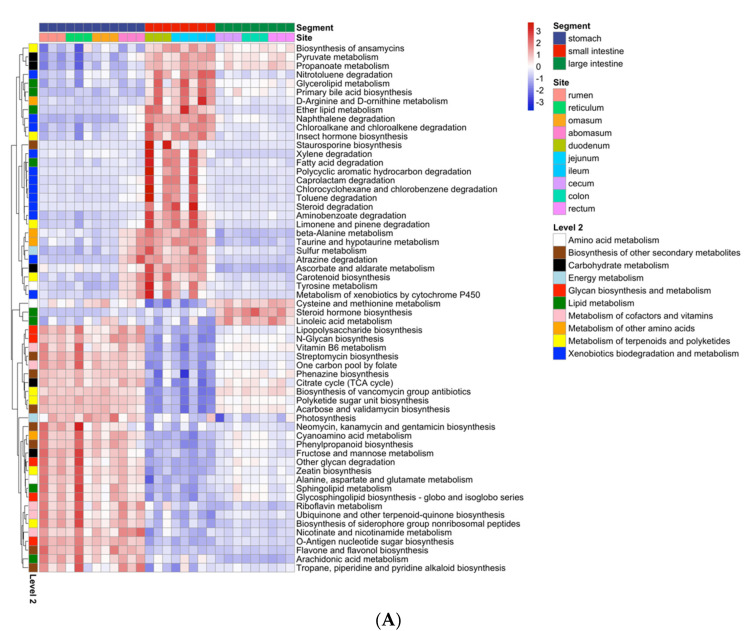
Characteristic KEGG pathways of the stomach, small intestine, and large intestine. (**A**) Heatmap of differentially enriched pathways related to the metabolism. The relative abundance of pathways was transformed into a Z-score, and second level categories of KEGG pathways are annotated with discrete colors in the left bar of the plot. (**B**) The relative abundance and logarithmic LDA score of characteristic KEGG pathways in other first level categories, rather than in metabolism.

## Data Availability

The data that support the findings of this study are available from the corresponding author, Y.W., upon reasonable request.

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
