# Peer review of "Characterization of the Microbial Communities along the Gastrointestinal Tract in Crossbred Cattle"

_animals, 2022, doi:10.3390/ani12070825_

Round 1
Reviewer 1 Report
Manuscript animals-1630677, entitled “Characterization of the microbial communities along the gastrointestinal tract in crossbred cattle”
Recommendation: The above paper is not suitable for publication in its present form.
General comment
The article provides useful information about the characterization of the microbial communities along the gastrointestinal tract in crossbred cattle. Although, the experiment was in general appropriately designed and implemented, there are some points that should be corrected or clarified.
L13-14: “Crossbreeding has been worldwide used to improve milk production, milk composition, and reproduction performance. Understanding…”
L16: “…technologies, with emphasis on nutrition and sustainability. In this…”
L18: “abundance” instead of “richness”
L20: “specific” instead of “corresponding”
L22: “…bred, and specific microbial biomarkers were identified in different regions.”
L23: “Gastrointestinal microbiota greatly affect health status and production performance in bovine.”
L26: “is not thoroughly” instead of “was less”
L31: “categorized” instead of “separated”
L61: “previous” instead of “past”
L61-62: Please check comment in L13-14
L65: “these crossbred cattle” instead of “this bovine”
L66: “with emphasis on” instead of “particularly in”
L67: Please delete “systems”
L70: “not thoroughly” instead of “less”
L78: “…crossbred cattle, and to provide useful information for the significance of gastrointestinal…”
L239-240: “…which may a result of GIT intestine physiological factors, such as oxygen…”
L244: “…proving that microbiota…”
L247: Where is this finding shown?
L247-248: “Among Firmicutes, Ruminococcaceae, Lachnospiraceae and Christensenellaceae…”
L249: “…the whole GIT, a finding that is consistent with that of a study conducted in sheep…”
L254: “…large intestine in the present but also in previous…”
L257: “…have the ability…”
L261: “…cellulose has been previously confirmed [34,35].”
L263: “…about the two-thirds...”
L270: “exerts” instead of “presents”
L271: Please rephrase “are lacking in the host”
L272: “However, a lot of bacteria…”
L278: “…with a previous study…”
L279-280: “and different GIT regions were characterized by…”
L282: “…confirmed that some…”
L284: “…by the enrichment…”
L293: “categorized” instead of “separated”
L295: “specific” instead of “corresponding”
Author Response
Dear Reviewer:
Thank you for all your valuable revision and suggestions, which greatly improved the quality of our manuscript. We have revised our manuscript according to all of your comments. There are two comments which we should explain to you.
#1: L18: “abundance” instead of “richness”
AU: Thank you for the suggestion. The word “abundance” may be inappropriate here since we use “richness” to describe how many kinds of bacteria/bacteria functional genes existed in one sample. However, the meaning of “richness” may be ambiguous here, so we replaced "richness" with "diversity".
#2: L247: Where is this finding shown?
AU: As is shown in Figure 2A, phylum Firmicutes was the second most abundant in stomach and the most abundant in small intestine and large intestine. L247 is now L272 in the newly revised manuscript.
Reviewer 2 Report
Dear authors It is not clear from the text whether the distribution of these bacterial counts has been checked whether or not they have a normal distribution.And why use the Wilcoxon rank sum test instead of the unpaired and paired Student's t-tests (also known as “t-test for matched pairs” or “t-test for dependent samples”?
Author Response
Dear Reviewer:
Thank you for your valuable comment.
#1: Dear authors It is not clear from the text whether the distribution of these bacterial counts has been checked whether or not they have a normal distribution.
And why use the Wilcoxon rank sum test instead of the unpaired and paired Student's t-tests (also known as “t-test for matched pairs” or “t-test for dependent samples”?
AU: Thank you for your valuable comments. We employed non-parametric tests because of the nature of microbiome data. As referred in the reference [1] list below, counts of bacteria in most cases violate the main assumption of typical parametric tests (normal distribution), whereas non-parametric tests are much more robust to the underlying distribution of the data since they are distribution-free approaches. We have also checked the distribution of our data, (relative) abundance and alpha diversity indices is not normal distributed. Furthermore, non-parametric test is more appropriate for small data size.
Reference:
[1] Segata, N., Izard, J., Waldron, L. et al. Metagenomic biomarker discovery and explanation. Genome Biol 12, R60 (2011). https://doi.org/10.1186/gb-2011-12-6-r60
Reviewer 3 Report
Dear Authors
In this study, the gastrointestinal microbiota of Simmental×Holstein crossbred cattle using 16s rRNA gene sequencing were analysed. Microbial communities, and three groups of GIT regions, including stomach, small intestine, and large intestine were characterized by corresponding bacteria and bacterial functions. Results showed that spatial heterogeneity of microbiota was found across the GIT of crossbred, and identified microbial biomarkers of different regions. This is an innovate study since there is no literature investigating microbial communities across the whole GIT in crossbred cattle.
The manuscript is within the journal's scope. Title is clear and informative; it displays the main objective of the study. The abstract is sectioned. It contains focused background with clear objective. The Introduction is concise. The figures and graphs show adequately the important results.
Minor corrections
In Material and methods, the small sample size needs to be explained to be better explained. How was the cross contamination avoided during the sample collection? It is not explicit from the manuscript.
In Discussion include information about the importance of the phylum Firmicutes.
Author Response
Dear Reviewer:
Thank you for all your valuable suggestions, which greatly improved the quality of our manuscript. We have revised our manuscript according to your comments.
#1: In Material and methods, the small sample size needs to be better explained. How was the cross contamination avoided during the sample collection? It is not explicit from the manuscript.
AU: In order to avoid cross-contamination during sample collection, plates for transferring GIT segments were only used once for each segment, and new medical gloves were used for each sampling. We have revised material and methods in our manuscript.
#2: In Discussion include information about the importance of the phylum Firmicutes.
AU: Thank you for your valuable suggestions, we have included information about the importance of the phylum Firmicutes in the manuscript.